# The Orexigenic Force of Olfactory Palatable Food Cues in Rats

**DOI:** 10.3390/nu13093101

**Published:** 2021-09-03

**Authors:** Fiona Peris-Sampedro, Iris Stoltenborg, Marie V. Le May, Pol Sole-Navais, Roger A. H. Adan, Suzanne L. Dickson

**Affiliations:** 1Department of Physiology/Endocrine, Institute of Neuroscience and Physiology, The Sahlgrenska Academy at the University of Gothenburg, 40530 Gothenburg, Sweden; iris.stoltenborg@gu.se (I.S.); marie.lemay@neuro.gu.se (M.V.L.M.); r.a.h.adan@umcutrecht.nl (R.A.H.A.); 2Department of Obstetrics and Gynaecology, The Sahlgrenska Academy at the University of Gothenburg, 40530 Gothenburg, Sweden; pol.sole.navais@gu.se; 3Brain Center Rudolf Magnus, Department of Translational Neuroscience, University Medical Center Utrecht, Utrecht University, 3584 Utrecht, The Netherlands

**Keywords:** food cues, feeding, food-seeking, ghrelin, arcuate nucleus, AgRP, POMC, dopamine, Fos, RNAscope

## Abstract

Environmental cues recalling palatable foods motivate eating beyond metabolic need, yet the timing of this response and whether it can develop towards a less palatable but readily available food remain elusive. Increasing evidence indicates that external stimuli in the olfactory modality communicate with the major hub in the feeding neurocircuitry, namely the hypothalamic arcuate nucleus (Arc), but the neural substrates involved have been only partially uncovered. By means of a home-cage hidden palatable food paradigm, aiming to mimic ubiquitous exposure to olfactory food cues in Western societies, we investigated whether the latter could drive the overeating of plain chow in non-food-deprived male rats and explored the neural mechanisms involved, including the possible engagement of the orexigenic ghrelin system. The olfactory detection of a familiar, palatable food impacted upon meal patterns, by increasing meal frequency, to cause the persistent overconsumption of chow. In line with the orexigenic response observed, sensing the palatable food in the environment stimulated food-seeking and risk-taking behavior, which are intrinsic components of food acquisition, and caused active ghrelin release. Our results suggest that olfactory food cues recruited intermingled populations of cells embedded within the feeding circuitry within the Arc, including, notably, those containing the ghrelin receptor. These data demonstrate the leverage of ubiquitous food cues, not only for palatable food searching, but also to powerfully drive food consumption in ways that resonate with heightened hunger, for which the orexigenic ghrelin system is implicated.

## 1. Introduction

Food cues in our environment are powerful triggers that encourage us to search for and consume food, even when there is no physiological need for it, thus standing as an important risk factor for obesity [1,2,3]. Most studies exploring the mechanisms underpinning cue-potentiated feeding use classical Pavlovian conditioning models, in which animals learn through association that a nonfood stimulus (e.g., either discrete (a light or a tone) or contextual cues) signals food availability [4,5,6,7,8,9,10]. However, in nature, animals/humans mostly rely on the sense of smell or sight to perceive environmental food stimuli.

It is well-known that the sense of smell is a central driver of food-seeking, appetite, and food preference in vertebrates, including humans [11,12], but one of those lingering questions in physiology research concerns the identification of the neural substrates involved in these feeding related outcomes. Classical neuroanatomical and electrophysiological studies indicate the existence of an olfactory–hypothalamic axis of potential relevance for feeding control [13], and connections between the main olfactory bulb and the hypothalamic arcuate nucleus (Arc; a core hub in the feeding neurocircuitry) have been described [14,15]. Recent advances using real-time fiber photometry recordings demonstrate the engagement of orexigenic agouti related protein (AgRP) and anorectic proopiomelanocortin (POMC) neurons, key feeding regulators residing in the Arc, in almost instantly sensing food availability in the olfactory modality prior to its consumption [16]. However, an important unanswered question is whether there are other relevant cell populations in this area responding to olfactory food cues under a non-food-deprived situation.

As is the case for other brain areas involved in feeding control, olfactory circuits express receptors for various circulating metabolic hormones. One prominent example is the receptor for the orexigenic hormone ghrelin (the growth hormone secretagogue receptor, GHSR) [12,17]. Indeed, the ghrelin system emerges as a possible link between olfactory cue-driven appetite and heightened food intake. Moreover, the possibility exists that external orexigenic signals originating from food cues in the environment (e.g., the smell of food) and from intrinsic hunger signals (e.g., ghrelin) converge on overlapping circuits to impact on feeding behaviors—not only food-seeking and intake, but also, as previously demonstrated, on pathways involved in exploratory sniffing and olfactory sensitivity [18].

In this study, we sought to discover whether a sustained exposure to an olfactory palatable food-evoking cue could increase the intake of a less palatable but readily available food, such as regular chow, in non-food-deprived rats subjected to a home-cage, hidden palatable food paradigm [19]. As the timing of the cue potentiated feeding effect, if any, remains elusive, we likewise investigated whether it is transient, or whether it persists, and for how long, upon cue presentation. To gain an understanding of the degree of crosstalk between internal metabolic and external olfactory food-evoking signals in non-food-deprived conditions, we explored the molecular identity of neurons in the Arc that are activated upon exposure to this paradigm. We hypothesized that ghrelin-responsive neurons are activated upon sensing the palatable food, even when food, whatever its nature, is not available for consumption. 

## 2. Materials and Methods

### 2.1. Animals and Experimental Model

Male Sprague-Dawley rats (*n* = 88), aged 8/10 weeks old upon arrival, were purchased from Charles River (Calco, Italy). The rats weighed 322–406 g on the first day of experimentation. Male C57BL/6N mice (*n* = 10), aged 12 weeks old and weighing 30.3 ± 1.1 g on the first experimental day, were also purchased from Charles River. Animals were allowed one-week acclimatization to the facility prior to being single-housed. They had *ad libitum* access to water and chow (2016 Teklad diet, Harlan Laboratories, Cambridgeshire, UK), unless otherwise stated, and were kept under standard conditions of temperature (20 ± 2 °C) and humidity (50 ± 10%) on a 12 h light–dark cycle (lights on at 7:00 a.m.) for the duration of the studies. We used a behavioral preparation [19], in which a metallic, perforated, opaque tea-strainer ball was suspended from each animal’s cage lid prior to single housing to avoid neophobia (to ensure that the animals were familiar with the set up before each experiment started) and to prevent it from becoming a non-food related contextual cue [20]. During experiments, this device contained the inaccessible palatable food (peanut butter, PB; Skippy, Hormel foods, Austin, MN, USA), delivering an olfactory cue to the home environment. To avoid cross-odor contamination, experiments in PB taste-naïve rats were carried out first (and on different days) from those in PB taste-familiar rats.

All experiments were approved by the local ethics committee for animal care in Gothenburg (ethics #132-2016) and complied according to European guidelines (Decree 86/609/EEC).

### 2.2. Feeding Response and Meal Patterns in an Olfactory PB Cue-Enriched Environment

An automated feeding monitoring system (TSE LabMaster, Project 4261, TSE Systems, Bad Homburg, Germany) was used to analyze feeding patterns (measuring cumulative chow intake [g], meal frequency, meal size [g], and ingestion rate [g/min]) in an olfactory PB cue-enriched environment in both PB taste-naïve (*n* = 16) and PB taste-familiar (*n* = 16) rats. This system allows uninterrupted and undisturbed recording of individual meals for each animal. Food hoppers containing regular pelleted chow were suspended on sensors (calibrated prior to starting the experiments) recording food intake to a sensitivity of 1 mg. Rats were transferred into the cages and allowed one-week habituation prior to starting the feeding recordings.

#### 2.2.1. Novel Olfactory PB-Cue Stimulus: PB Taste Naïve Rats

First, we generated PB taste-naïve rats (*n* = 16). To keep the familiarization procedure as similar as possible between PB taste-naïve and PB taste-familiar groups, rats were presented with an inedible object (empty food dish) once a day for three alternating days after habituation to the experimental cage. To avoid interval-dependent anticipation, the object was presented at different time points and no identical intervals were repeated. Afterwards, baseline chow intake was calculated based on the daily automatically recorded feeding measurements. Data were averaged from the last two days of measurements and reported as 1, 4, 8, 12, 16, 20 and 24 h chow intake. Afterwards, open tubes containing PB were encased in the perforated balls hanging from each of the lids immediately after the light cycle onset and left there for 24 h. Cumulative chow consumption in the olfactory cue-enriched environment (1, 4, 8, 12, 16, 20 and 24 h post-cue introduction), as well as within 24 h following cue removal, was recorded and ultimately compared to that of the baseline period.

#### 2.2.2. Conditioned Olfactory PB-Cue Stimulus: PB Taste-Familiar Rats

Two weeks later, another cohort of rats (*n* = 16) were exposed to PB tasting. Briefly, they were subjected to the same familiarization protocol as the PB taste naïve rats, but they were given 1-h access to consume a limited amount of PB for three alternating days, at varying time intervals, instead of the inedible object. The amount of PB consumed was measured after each tasting session. After baseline food measurements, PB-filled tubes were encased in the perforated balls, as described above. Cumulative chow consumption in the olfactory cue-enriched environment (1, 4, 8, 12, 16, 20 and 24 h post-cue introduction), as well as within 24 h following cue removal, was recorded and ultimately compared to that of the baseline period.

#### 2.2.3. Meal Pattern Analysis in the Novel and the Conditioned Settings

Data for meal pattern analysis were collected as binary data every 10 s. Meal analysis was undertaken using LabMaster software (TSE Systems), whereby all meals occurring during the study period (baseline and 24-h exposure to the cue enriched environment in both PB taste-naïve and PB taste-familiar groups) were recorded chronologically to allow the evaluation of single feeding bouts. Meals were defined as an episode of feeding in which at least 0.5 g of chow was removed, with meal termination criterion as the beginning of at least 10-min pause in ingestion [21]. Ingestion rate during a meal was calculated by dividing meal size by meal duration. Meal frequency and meal size were summarized over different periods: light phase (12 h), dark phase (12 h), and total day (24 h), and then averaged per rat and group.

#### 2.2.4. Cue-Induced Feeding Response in PB Taste-Familiar Mice

Mice (*n* = 10) were first familiarized with PB taste as described for PB taste-familiar rats. The amount of PB consumed was, likewise, measured after each tasting session. A tube containing PB was encased in each of the perforated balls and chow intake was measured manually (using scales that had a precision of 1 mg) at 1, 3 and 6 h post-cue introduction. The same time points were used for the control condition on the day before, when we measured spontaneous food intake in the absence of the olfactory PB cue following introduction of an empty tube in the perforated ball.

### 2.3. Food-Seeking in Familiar and Risky Environments Enriched with an Olfactory PB Cue

We explored ways to quantify the dramatic food-seeking behavior observed especially in PB taste-familiar rats upon introduction of the olfactory PB cue in the home cage, which persisted for around 30 min (Appendix A). PB taste-naïve (*n* = 15) and PB-taste familiar (*n* = 20) rats were used.

#### 2.3.1. Acoustic Measurements in the Home Environment

All rats were initially brought to the experimental room 30 min prior to starting the recordings for acclimatization. All the recordings were carried out using the Sound Level Meter app [22] and took place early in the morning (~8:00 a.m.), to avoid the bustle of the facility, and at the same time for each cohort. After recording the baseline noise (minimum, average and maximum intensities [dB]) following introduction of an empty tube in each of the perforated balls, the latter was replaced with a PB-filled tube and the environmental noise was then recorded (minimum, average and maximum intensities [dB]) in the presence of PB odor. The sound intensity difference (∆dB) between the baseline and the cue-setting recordings was calculated for the total 15-min period.

#### 2.3.2. PB-Baited Open Field Test

Following acoustic measurements, rats from each cohort were subdivided into four subgroups (PB taste-naïve/PB cue (*n* = 7), PB taste-naïve/No cue (*n* = 8), PB taste-familiar/PB cue (*n* = 10) and PB-taste familiar/No cue (*n* = 10)) and subjected to a PB-baited open field test (adapted from [23]), which was used as a conflict-based approach to assess risk-taking behavior in food-seeking. The open field boxes consisted of two different bright white 900 × 900 mm arenas, protected with 300 mm-high walls (Med Associates Inc., St Albans, VT, USA). On the day of the experiment, either an empty or a PB-containing perforated ball was placed in the center of the field and secured to the floor. Each rat was initially placed in the left corner of the field and allowed to freely explore the space without interruption for 10 min. Each trial was filmed for later video analysis, which was carried out manually. To ease the analysis, we draw a virtual “cue zone” (the area immediately surrounding the perforated ball), which was kept constant for all the trials. Latency was defined as time to approach the perforated ball after introduction into the arena. The duration of the first contact with the perforated ball, as well as the total time spent in contact with the set up, was measured. The perforated ball was cleaned with ethanol between each trial. The behavioral assessment took place during the light phase (10:00 a.m. to 14:00 p.m.) and groups were counterbalanced regarding the time of testing.

### 2.4. Cell Activation in the Arcuate Nucleus upon Olfactory Detection of PB

PB taste-naïve (*n* = 15) and PB taste-familiar (*n* = 13) rats were used to explore whether the olfactory PB cue activated cells in the Arc in non-food-deprived conditions. *Ad libitum*-fed rats were exposed to either an olfactory PB cue-enriched environment (PB taste-naïve/PB cue, *n* = 8; PB taste-familiar/PB cue, *n* = 8) or to a non-enriched environment (PB taste-naïve/No cue, *n* = 7; PB taste-familiar/No cue, *n* = 5) for 20 min. Food access was withheld from cue exposure until sacrifice. At 80 min from cue removal, rats were deeply anesthetized with a mixture of Rompun vet.^®^ (10 mg/kg; Bayer, Leverkusen, Germany) and Ketaminol vet.^®^ (75 mg/kg; Intervet, Boxmeer, The Netherlands), prior to being perfused transcardially with heparinized 0.9% saline followed by 4% paraformaldehyde (PFA) in 0.1 M phosphate buffered saline (PBS), following a group counterbalanced timing. After harvesting, the brains were stored overnight at 4 °C in a 4% PFA fixative solution containing 15% sucrose, followed by a minimum of 12-h incubation in 30% sucrose/0.1 M PBS to ensure cryoprotection. Coronal sections (30-μm thick) containing the Arc were cut using a cryostat and stored in an antifreeze solution (25% glycerin, 25% ethylene glycol, 50% 0.1 M PBS) at −20 °C until further processing.

Free-floating sections were processed for the immunohistochemical detection of Fos protein using the 3,3′-diaminobenzidine (DAB)-hydrogen peroxidase method [24]. After deactivation of endogenous peroxidases, the sections were rinsed with 0.1 M PBS + 0.3% Triton X-100 prior to being blocked for 1 h at room temperature in 0.1 M PBS, 3% normal goat serum, 0.25% BSA and 0.3% Triton X-100. Afterwards, they were incubated with an anti-c-Fos rabbit primary antibody (dilution 1:20,000; Ab-5 (4-17) Rabbit pAb, PC38; Calbiochem, San Diego, CA, USA) for three nights. The sections were then rinsed and subsequently incubated for 2 h with a peroxidase goat antirabbit immunoglobulin (Ig)G secondary antibody (dilution 1:200; PI1000; Vector Laboratories, Burlingame, CA, USA) and a DAB, nickel, and hydrogen peroxide solution. Brain sections were mounted onto glass slides and coverslipped with ProLong^®^ Diamond Antifade mountant (Thermo Fisher, Waltham, MA, USA).

Unilateral images (5 sections per rat from 2.04 to 3.72 caudal to bregma) were acquired from rostral to caudal using a DMRB fluorescence microscope (10X/N.A. 0.30; Leica Microsystems, Wetzlar, Germany), as previously described in [24]. The number of Fos+ cells per section was counted manually in ImageJ/Fiji (NIH, Bethesda, MD, USA) using the Cell counter plug-in. The mean number of Fos+ cells per hemisection (and averaged per three blind countings) was calculated, averaged for each brain and, ultimately, for each experimental group.

### 2.5. Neurochemical Identification of the Cells Activated by the Olfactory PB Cue

Candidate Arc cells activated during exposure to the olfactory PB cue included those whose known distribution (based on the Allen Brain Atlas) shows some overlap with the Fos+ cells and that are known to be embedded within the feeding circuits (AgRP, POMC and dopamine) and/or that contain GHSR [16,25,26]. Triple fluorescent *in situ* hybridization using RNAscope^®^ [24] was performed to study the potential co-expression of Fos (*c-Fos* probed) with GHSR (*Ghsr* probed), AgRP (*Agrp* probed), POMC (*Pomc* probed) and dopamine (*Tyrosine hydroxylase* [*Th*] probed) in the Arc of PB taste-familiar rats (*n* = 3–4) upon a 20-min exposure to PB odor. To this end, three independent assays were run: (i) *c-Fos*, *Ghsr* and *Agrp*; (ii) *c-Fos*, *Ghsr* and *Pomc*; and (iii) *c-Fos*, *Ghsr* and *Th*. This also allowed us to study the colocalization of GHSR with AgRP, POMC and TH. Rats were deeply anesthetized with the mixture of Rompun vet.^®^ and Ketaminol vet.^®^ and perfused transcardially, as described above. After harvesting, the brains were stored overnight at 4 °C in a 4% PFA fixative solution, and then kept in 0.1 M PBS containing 25% sucrose at 4 °C until cryosection. Coronal sections containing the Arc (14-μm thick, every 6th section collected to provide six adjacent series) were cut using a cryostat and stored in an antifreeze solution (25% glycerin, 25% ethylene glycol, 50% 0.1 M autoclaved PBS) at −20 °C until further processing.

All reagents were purchased from Advanced Cell Diagnostics (ACD, Hayward, CA, USA), if not stated otherwise. The *c-Fos* probe (#403591-C3) contained 20 oligonucleotide pairs and targeted region 473-1497 (Acc. No. NM_022197.2) of the *c-Fos* transcript. The *Ghsr* probe (#480031) contained 14 oligonucleotide pairs and targeted region 2-742 (Acc. No. NM_032075.3) of the *Ghsr* transcript. The *Agrp* probe (#316171-C2) contained 13 oligonucleotide pairs and targeted region 14-613 (Acc. No. NM_033650.1) of the *Agrp* transcript. The *Pomc* probe (#318511-C2) contained 17 oligonucleotide pairs and targeted region 21-921 (Acc. No. NM_139326.2) of the *Pomc* transcript. The *Th* probe (#314651-C2) contained 20 oligonucleotide pairs and targeted region 422-1403 (Acc. No. NM_012740.3) of the *Th* transcript. Negative and positive control probes were processed in parallel with the target probes to ensure RNA integrity and an optimal assay performance. On the day prior to the assay, the sections were mounted onto SuperFrost Plus slides (#631-9483; VWR, Radnor, PA, USA) and baked at 60 °C overnight in a HybEz oven (#321462). On the day of the assay, slides were first incubated for 7 min in hydrogen peroxide (#322335), submerged in Target Retrieval buffer (#322001) and rinsed in autoclaved Milli-Q purified water. The slides were quickly dehydrated in 100% ethanol and allowed to air-dry. All the sections were then incubated with Protease Plus (#322331) for 30 min. The subsequent steps were performed according to the manufacturer’s protocol for the tyramide-based RNAscope^®^ Multiplex Fluorescent v2 Assay (#323100). The *Agrp*, *Pomc* and *Th* probes were labeled with Opal 520 (1:500; FP1487A; PerkinElmer, Waltham, MA, USA), the *Ghsr* probe with Cy3 (1:2000; Akoya Biosciences, Menlo Park, CA, USA), and the *c-Fos* probe with Cy5 (1:3000; Akoya Biosciences). All the sections were counterstained with DAPI, coverslipped with ProLong^®^ Diamond Antifade mountant (Thermo Fisher) and stored in the dark at 4 °C until imaging.

Images for the quantification of RNAscope data were captured using a laser scanning confocal microscope (LSM 700 inverted, Zeiss, Oberkochen, Germany) equipped with a Plan-Apochromat 40x/1.3 Oil DIC objective (used at the Centre for Cellular Imaging at Gothenburg University). Tile scans (5 × 5, 5 × 6 or 5 × 7 tiling, depending on the bregma) and Z-stacks (optical section of 1.0 μm) of the Arc-containing sections were captured unilaterally from rostral to caudal (2–4 sections per rat, from 2.04 to 3.48 mm caudal to bregma). Laser intensities for the different channels were kept constant throughout the imaging process. The Z-stack images were processed using the maximum intensity projection function in the Zen Black software (Zeiss). The final images were then stitched and the cells counted in ImageJ/Fiji (NIH). The Cell counter plug-in was used to count positive cells and colocalization in the Arc. DAPI stain was used for cellular recognition. DAPI-identified cells with >1–3 dots/cell were defined as being positive for a given peptide [24]. The quantification of the co-expression per hemisection was averaged for each brain and, ultimately, for each experimental group.

### 2.6. Assessment of Active Ghrelin Levels upon Olfactory Detection of PB

Another cohort of PB taste-familiar rats (*n* = 21) was used to confirm the cue-induced hyperphagic effect. After habituation to PB taste, PB was encapsulated in the perforated balls and the intake of chow measured manually at 1, 3 and 6 h post-cue introduction. The same time points were used as for the control condition on the day before, in which spontaneous food intake was measured in the absence of PB odor. The same rats were re-exposed to either an olfactory PB cue-enriched environment (*n* = 13) or to a non-enriched environment (perforated balls with empty tubes; *n* = 8) for 1 h, after which they were anesthetized and sacrificed to obtain blood for active ghrelin measurement. Food was only withheld during the 1-h exposure to the olfactory cue. Following anesthesia with isoflurane and decapitation, trunk blood was immediately collected into EDTA-coated tubes containing 4-(2-Aminoethyl)benzenesulfonyl fluoride hydrochloride (AEBSF) to a final concentration of 1 mg/mL. We did not acidify the sample because it has been shown that HCl addition to AEBSF-treated samples does not provide enhanced hormone stability [27]. To avoid cross-odor contamination, rats from different groups were kept in adjacent rooms. Sacrifices took place from 10:00 a.m. to 12:00 a.m. and groups were counterbalanced with respect to the time of sacrifice. Tubes were ultimately centrifuged to obtain plasma, which was aliquoted and stored at −80 °C until processing. Plasma acyl-ghrelin levels were measured in duplicate using a commercial ELISA kit (#EZRGRA-90K; Merck Millipore, Darmstadt, Germany) following the manufacturer’s instructions. Samples were thawed only once.

### 2.7. Statistical Analysis

The program IBM SPSS Statistics 27 (IBM Corp., Armonk, NY, USA) was used for statistical analyses. Comparisons were carried out by one-way repeated measures ANOVA when assessing the automated feeding response and meal patterns upon exposure to the olfactory PB cue in rats (with *Conditioning* (naïve, familiar) as “between factor” and *Cue* (cue absent, olfactory PB cue, cue removal—when applicable) as “within factor” variables). We used paired samples *t*-tests to compare the cue-induced feeding response to that of the baseline manual measurements in both PB taste-familiar rats and mice, as well as the intake of PB during the different taste conditionings in PB taste-familiar rats and mice, and a one-way ANOVA (*Cue*) to assess the effects of the olfactory cue on active ghrelin levels in PB taste-familiar rats. All other data were subjected to two-way ANOVA analyses (*Conditioning* × *Cue*). 

Given that we did not contemplate an explicit comparison between PB taste-naïve and familiar rats, additional paired samples *t*-tests or one-way ANOVAs (*Cue*) were used to follow up significant main effects and/or interactions. 

Scatterplots and boxplots were generated using R (version 3.6.3, ggplot2 package) and other plots express the mean ± SEM. Statistical significance was set at *p* < 0.05, and values 0.05 ≤ *p* < 0.1 were considered evidence of statistical trends. Statistical annotations of the main analysis include the *p* value and its corresponding F or t ratio together with the degrees of freedom.

## 3. Results

### 3.1. Olfactory Detection of PB Increases Meal Frequency to Cause Chow Overconsumption in Non-Food-Deprived PB Taste-Familiar Rats

To determine whether olfactory detection of PB can trigger overconsumption of chow in non-food-deprived rats, PB taste-naïve and PB taste-familiar rats were subjected to automatic feeding recordings in the presence of a PB odor for 24 h (Figure 1A). 

The statistic output revealed an overall effect of the cue intervention to increase chow intake 8, 12, 16 and 20 h post-cue introduction in the home cage (*Cue*: F_[1,28]_ = 8.315, *p* = 0.007; F_[1,28]_ = 19.076, *p* < 0.001; F_[1,28]_ = 6.142, *p* = 0.019; and F_[1,28]_ = 3.211, *p* = 0.083, respectively), and a *Conditioning* × *Cue* interaction for the 1 h measurement (F_[1,28]_ = 6.452, *p* = 0.016). Strikingly, PB taste-familiar rats showed a sustained overconsumption of chow in the presence of the olfactory PB cue that lasted up to 16 h (baseline vs. cue exposure: 1 h, *p* = 0.019; 8 h, *p* = 0.025; 12 h, *p* = 0.005; and 16 h, *p* = 0.022 (not significant for the 4-h measurement: *p* = 0.248)) (Figure 1B). Notably, rats that had never tasted PB before did not change their food intake despite the presence of the PB odor, suggesting that the PB did not serve as a food cue in this group (Figure 1B). Cumulative 24-h chow intake following cue removal was similar to that of the baseline and the cue settings in both the PB taste-naïve and familiar rats (Appendix A).

Consistent with the stimulatory effect of the PB-linked olfactory cue on chow intake, meal patterning was, likewise, affected (*Cue* [meal size]: F_[1,28]_ = 4.164, *p* = 0.050; *Conditioning* × *Cue* [meal frequency]: F_[1,28]_ = 21.953, *p* < 0.001). PB taste-naïve rats tended to increase the number of meals during the light phase upon exposure to the olfactory cue (baseline vs. cue exposure: *p* = 0.058), although this effect was ultimately offset, since the same analysis confirmed reductions in the number of meal bouts during both the dark phase and the entire 24-h period (baseline vs. cue exposure: *p* < 0.001 and *p* = 0.006, respectively) (Figure 1C). PB taste-naïve rats consumed, therefore, fewer meals throughout exposure to PB odor, which might reflect a hypophagic effect linked to the introduction of a novel odor in the cage. They, nonetheless, compensated their lower meal frequency by increasing meal size overall (baseline vs. cue exposure, dark phase: *p* = 0.006; entire 24-h period, *p* = 0.050) (Figure 1D). PB taste-familiar rats, on the contrary, ate more frequently when exposed to the olfactory cue at all the time periods studied (baseline vs. cue exposure, light phase: *p* = 0.036; dark phase: *p* = 0.092; and entire 24-h period: *p* = 0.004) (Figure 1C). Interestingly, such an increase in meal frequency in this group did not cause a compensatory reduction in meal size (Figure 1D). The ingestion rate was not significantly affected by the presence of the olfactory cue in any of the two groups (baseline vs. cue exposure, PB taste familiar rats: 1.52 ± 0.24 g/min and 1.64 ± 0.40 g/min, respectively; PB taste naïve rats: 2.35 ± 0.40 g/min and 1.93 ± 0.39 g/min, respectively).

In a parallel study, a cohort of PB taste-familiar mice was used to determine whether the olfactory detection of PB could, likewise, influence chow intake in mice. In agreement with Boone and coworkers [29], the olfactory detection of a familiar palatable food did not influence chow consumption in mice (Appendix A), probably because of the small amount of food a mouse consumes over a day, especially during the light phase, at a time during which measurements took place.

As expected (if we consider a plausible neophobic response to a different taste and caloric content), PB consumption was greater during the second tasting session compared to the first one (rats: 87% increase, t_[15]_ = −6.299, *p* < 0.001; mice: 67% increase, t_[9]_ = −3.191, *p* = 0.011), and then plateaued during the last familiarization to PB taste (Appendix A). Hence, subjecting rodents to a single tasting experience of a palatable food does not seem optimal to foster its craving and subsequent seeking, when cued by an olfactory stimulus.

### 3.2. Olfactory Detection of PB Prompts Food-Seeking in Non-Food-Deprived PB Taste-Familiar Rats

Besides acoustic measurements in the home cage, which were higher in the PB taste-familiar than in the PB taste-naïve group (Appendix A), we also explored the incentive salience of PB in a risky environment in the same cohorts of PB taste-naïve and PB taste-familiar rats (Figure 2A).

Baiting the center of the arena with PB odor did not evoke food-seeking in PB taste-naïve rats (Figure 2B,C). The duration of the first contact with the perforated ball in the cue-setting did, on the contrary, increase significantly by 55% in rats that were familiar with PB taste (Cue: F_[1,30]_ = 3.921, *p* = 0.057; no cue vs. PB cue: *p* = 0.043) (Figure 2B). Likewise, PB taste-familiar rats spent 93% more time exploring the bait (in comparison with the time spent exploring the empty perforated ball) during the entire 10-min test, despite the aversive environment (Cue: F_[1,30]_ = 15.660, *p* < 0.001; no cue vs. PB cue: *p* < 0.001) (Figure 2C). We did not observe any differences in latency to approach the set up for the first time for either group (Figure 2D), suggesting that the anxiety-like response was similar among groups [23].

### 3.3. Olfactory Detection of PB Activates Cells in the Arcuate Nucleus in Non-Food-Deprived PB Taste Familiar Rats

To explore the neural substrates engaged in the orexigenic and food-seeking responses to the olfactory PB cue, we first examined whether it activated cells in the Arc. Non-food-deprived PB taste-naïve and PB taste-familiar rats were exposed to a cue-enriched environment or to a non-enriched environment for 20 min (Figure 3A).

Consistent with the food intake data, the olfactory PB cue induced Arc Fos expression only in PB taste-familiar rats (*Conditioning*: F_[1,27]_ = 3.934, *p* = 0.059; *Cue* × *Conditioning*: F_[1,27]_ = 6.697, *p* = 0.016; 148% more cells compared to the control condition, *p* = 0.047) (Figure 3B,C). Unexpectedly, we were unable to detect more than a few scattered Fos+ cells in any other hypothalamic or brain areas. The only exceptions were the piriform and cingulate cortex, the olfactory tubercle and the medial amygdaloid nucleus, all areas being important relays for olfactory information [31,32]. Fos was detected in these regions in all the groups exposed to the olfactory cue, independently of whether they were familiar with the PB taste or not (data not shown).

### 3.4. Intermingled and Overlapping Neuronal Populations in the Arc Are Activated upon Olfactory Detection of PB in Non-Food-Deprived Rats

We then sought to obtain a molecular census of the Fos+ cells in the Arc responding to the olfactory PB cue. The RNAscope study demonstrates that the olfactory detection of PB by *ad libitum*-fed rats does not merely activate a single Arc neuronal population, but rather intermingled populations of cells (Figure 4A, Figure 5A and Figure 6A). Of the distinct types of neurons analyzed, as many as 20.5 ± 0.9% of Fos+ cells co-expressed AgRP (Figure 4B), almost a quarter (23.5 ± 1.9%) were POMC+ (Figure 5B), while only 12.6 ± 1.0% were dopaminergic (i.e., expressed *Th* mRNA) (Figure 6B). Notably, 35.7 ± 2% of Fos+ cells expressed GHSR (averaged from all the data of the three assays (39.1, 34.9 and 32.1%); Figure 4C, Figure 5C and Figure 6C), which turned out to be the most prominent cluster. The proportion of GHSR, AgRP, POMC and TH+ cells that also expressed Fos was 24.8 ± 3.0 (averaged from all the data of the three assays for GHSR (31.6, 23.4 and 16.8%)), 24.2 ± 2.3, 36.1 ± 4.4 and 24.2 ± 6.5%, respectively (Figure 4B, Figure 5B and Figure 6B).

On the other hand, over half of the GHSR+ cells (56.5 ± 1.9%) co-expressed AgRP and 83.2 ± 4.0% of AgRP cells expressed GHSR (Figure 4D); 14.5 ± 1.8% of GHSR+ cells co-expressed POMC and, interestingly, approximately a third of POMC neurons (32.2 ± 4.2%) were GHSR+ (Figure 5D); and, finally, 20.1 ± 2.0% of GHSR+ cells expressed TH and two thirds of TH cells (68.0 ± 2.0%) expressed GHSR (Figure 6D). Figure 4E, Figure 5E and Figure 6E depict the molecular identities of the total population of cells expressing Fos for each RNAscope assay.

Regarding triple co-expressions, a quarter of the total population of cells (25.3 ± 3.1%) co-expressing AgRP and GHSR were recruited upon olfactory detection of PB (i.e., were Fos+, Figure 4F). The majority of the AgRP cells activated by the PB cue (87.9 ± 1.8%) also contained GHSR, and almost half of the GHSR cells activated by the PB cue (46.0 ± 1.7%) were also positive for AgRP (Figure 4F). More than a third of the total population of cells (39.4 ± 3.8%) co-expressing POMC and GHSR were recruited upon olfactory detection of PB (Figure 5F). Similarly, more than a third of the POMC cells activated by the PB cue (38.0 ± 8.8%) also contained GHSR; and nearly a third of the GHSR cells that were activated by the PB cue (28.6 ± 1.8%) were also positive for POMC (Figure 5F). Finally, approximately a quarter of the total population of cells (27.8 ± 7.3%) co-expressing TH and GHSR were recruited upon olfactory detection of PB (Figure 6F). Three quarters of the TH cells activated by the PB cue (77.9 ± 5.3%) contained also GHSR; and almost third of the GHSR cells activated by the PB cue (30.4 ± 2.2%) were also positive for TH (Figure 6F).

### 3.5. Olfactory Detection of PB Triggers Release of Active Ghrelin in Non-Food-Deprived PB Taste-Familiar Rats

After reproducing the effects of exposure to the PB smell on chow intake in a new cohort of PB taste-familiar rats fed *ad libitum* (baseline vs. cue exposure: 1 h, t_[20]_ = −2.951, *p* = 0.008; 6 h, t_[20]_ = −2.769, *p* = 0.012) (Figure 7A), we also measured their plasma levels of active (acylated) ghrelin, which were elevated by 33% upon olfactory detection of PB (*Cue*: F_[1,20]_ = 4.497, *p* = 0.047) (Figure 7B).

## 4. Discussion

We show that olfactory detection of a familiar palatable food affects feeding patterns, causing sustained overconsumption of the only available food option—here, plain chow—even if the latter lacks incentive value under *ad libitum*-fed conditions. In line with the orexigenic response observed, perceiving this cue in the environment leads towards a more risk-taking phenotype in food-seeking, and triggers the release of active ghrelin. Using Fos mapping, we identify the Arc as a key neural substrate activated by olfactory food cues, and our results suggest that these recruit intermingled populations of cells embedded within the feeding circuitry, even when neither chow nor the cued PB are available for consumption.

It is widely accepted that cues recalling food, such as its sight or smell, prompt food-seeking and initiate feeding despite satiation in humans. Satiated subjects, when exposed to a virtual food cue-rich environment, self-report increased hunger and an increased desire to eat, leading to a heightened motivation for food and increased caloric consumption [3]. The pioneering works of Weingarten, first, and Petrovich, more recently, have also demonstrated that both discrete and contextual cues potentiate feeding in rodents through classical Pavlovian conditioning [4,5,6,7,8,9,10]. In these settings, however, there exists a phase of associative learning that takes place in behavioral chambers and the ultimate feeding response is specific to the signaled food. Our results show that olfactory detection of a palatable food in the home cage is powerful enough to cause chow to be overconsumed, a less preferred food that lacks incentive value when not hungry. Boggiano and colleagues [20] previously demonstrated that context cues concomitantly present whilst consuming a palatable food (e.g., a bedding that differs from the animal’s habitual cage) become drivers to promote the overeating of chow when sated rats are introduced in this food-paired context. The perforated ball in our study was already present in the home cage prior to single-housing, and all the baseline food intake measurements took place after the PB tastings, in such a way that any contextual cues (i.e., the sudden presence of a new object in the cage or associative contextualization of the home cage with PB intake) could not have influenced the results. In a nutshell, our behavioral approach [19] allowed us not only to investigate long-term feeding outcomes in response to food-linked olfactory cues with small disturbance to the animals, but also to selectively isolate the cue-dependent effects from the associative learning context, which, *per se*, influence food intake [20].

The cue-driven orexigenic response was already detected after 1 h, suggesting that it develops rapidly upon cue presentation, but spanned over 16 h, indicating that it persists even long after the PB smell is detected for the first time. Questions that inevitably arise are: what drives rats in our model to respond to the PB cue by overeating chow, and why is this response sustained over time? Petrovich and colleagues, in their work, discussed the fact that the cue-driven orexigenic response underlies a specific motivation for the signaled food, akin to craving [6]. Our results suggest that the sensory detection of PB in the environment drives a general desire to eat, similar to that expected from experiencing hunger. The motivation for initiating a meal certainly included an early stage of specific craving for PB, which conceivably preceded the assimilation by the rats that the olfactory cue did not signal its access. Without further expectations of obtaining PB, the rats’ physiology may have worked towards fulfilling this new set point by encouraging the rats to overconsume the available chow. The fact that chow intake was not greater after 24 h, however, suggests that the animals eventually compensate for their earlier cue-linked overconsumption, and that energy homeostasis prevails, as already postulated decades ago by Weingarten [10]. Importantly, however, this substantial intake of chow, albeit transient, was not compensated for within the 24 h after cue removal, because daily chow intake post-cue was similar to both the baseline and the cue settings.

This temporary overconsumption appears to reflect, especially, an increased meal frequency that was inadequately compensated for. Such a higher frequency of meals likely underscores an effect of olfactory food linked cues in selectively activating appetitive systems regulating meal initiation within hypothalamic areas, whereas a failure to decrease meal size would indicate the engagement of hindbrain pathways involved in satiation and meal termination [28]. Together with our finding that circulating active ghrelin levels are increased 1 h after the presentation of the cue, and consistent with a role of endogenous ghrelin as a physiological meal initiator [33,34], these data would substantiate an engagement of the ghrelin system in driving, at least acutely, the appetitive and orexigenic components of the behavioral response triggered by the cue. Corroborating this idea, evidence from human studies link an enhanced response to primary food-linked cues (i.e., food pictures) with heightened circulating ghrelin [30]. Our data, therefore, support the notion that there exists a feed-forward mechanism whereby olfactory food cues engage the ghrelin system by enhancing ghrelin release, which eventually heightens food-cue sensitivity (arguably by boosting olfactory acuity [18]), food-seeking and feeding, behaviors that are well-known to be driven by this hormone [23,25,35]. The level at which this feed-forward mechanism occurs is unclear. It is tempting to speculate that it might be fueled by the direct action of ghrelin via its receptor on olfactory processing centers, as well as through its positive and inhibitory actions on AgRP and POMC neurons, respectively [36]. Yet another possibility is that olfactory inputs, through olfactory circuits, communicate directly with central neurons to regulate both food-seeking and ingestion.

Due to the pronounced food-seeking for the cued PB, we investigated the potential of such olfactory cues to influence risk-taking in a PB-baited open field test. It seems clear that this appetitive state directs the animal to, and motivates initial contact with, the cued food: the PB bait in the center of the field shifted the individual’s behavioral choice towards a more risk-taking phenotype, because the animals spent twice as much time exploring the perforated ball despite the anxiogenic environment. Consistently, both fasting and exogenous ghrelin have been found to reset the perceived pressure on food supply, thus encouraging sated mice to forgo the fear of an open field in favor of the chance of getting food [23].

The ensemble of responses to the cued PB suggest that olfactory food cues engage different pathways involved in motivation, feeding and energy homeostasis. Consistent with this, we found that a 20-min exposure to PB smell activated cells in the Arc in *ad libitum*-fed rats, similar to that observed for other orexigenic stimuli, such as fasting [37] and ghrelin administration [24]. Notably, the olfactory PB cue did not induce Fos protein expression in any other hypothalamic or reward-linked areas. This should not be interpreted as suggesting that these areas are not activated, but, rather, that if they actually are, the response is not coupled to the induction of Fos protein expression. Although it would have been expected that exposure to cues evoking palatable foods increases the responsiveness of the reward circuitry to promote the seeking and the subsequent cue (specific)-potentiated feeding [38], the interpretation of the Fos results must be addressed cautiously. As previously highlighted by others [39,40], the lack of a Fos response in a specific cell population does not rule out the involvement of the hosting brain area in a functional circuit.

Interestingly, GHSR-containing cells were found to be the more prominent cluster from all the cell types activated upon exposure to the PB smell. Did the olfactory cue activate GHSR-containing cells or was it a downstream consequence of cue-induced ghrelin release? Based on the available information, it would be difficult to estimate the extent to which cue-induced ghrelin may have directly contributed to mediating the effects of the olfactory cue as a part of our hypothesized feed-forward mechanism.

Whereas AgRP and POMC neurons have been so far considered to be sensors of energy requirements, the work of Chen and colleagues [16] clearly shows that their activity can be rapidly modulated by the chemosensory detection of food. These observations pave the way towards a new concept in which food-seeking would be a higher order consequence of detecting food in the environment, rather than a downstream consequence of sensing an energy deficit. In line with this, our results suggest that the olfactory detection of palatable food in the environment activates both AgRP and POMC neurons to a similar extent. Interestingly, the ghrelin system seems to be an important liaison between hunger and responsiveness to olfactory cues: the majority of the recruited AgRP cells also expressed GHSR, while a substantial proportion of POMC did so too. It is worth recalling, at this juncture, that although Fos expression indicates a change in neuronal activity, it does not always reflect increased electrical activity [39]. Then, as introduced above, the triple co-expression results might, likewise, reflect a complex crosstalk between the ghrelin system and external olfactory food-evoking signals, since they might underlie a direct effect of cue-induced ghrelin on AgRP and POMC cells.

Although previous work has shown that dopaminergic cells in the Arc respond to fasting and peripheral ghrelin, and, thus, probably stand as a novel actor regulating energy homeostasis and food intake [26], our results suggest that these neuronal population might not be of critical relevance for integrating food-linked sensory stimuli in the olfactory modality.

It is worth emphasizing that only rats conditioned to the taste of PB showed food-cue reactivity, thereby displaying the cue-evoked motivational and feeding responses. Merely challenging the PB taste-naïve rats with PB smell was insufficient to initiate interest for the hidden food or boost chow intake, suggesting that the odor only gains value when the animals are familiar with the taste and ingestion. Although rodents might sense the intrinsic hedonic value of nutritive stimuli (e.g., fat or sugar), this might be not enough to condition a taste preference, and, thus, the postingestive actions of nutrients must take place [41], underpinning the role of learning and associative processes in cue-potentiated feeding [4,5,6,7,8,9,10].

## 5. Conclusions

Collectively, our data endorse a role of ubiquitous olfactory food stimuli not only in driving food-seeking (most probably directed towards the signaled food) and release of ghrelin, which appears to be a prominent and versatile actor, but also ultimate feeding, which develops towards the only food available, even if it is *a priori* non-salient. Our finding that populations of Arc cells embedded within the energy balance—hunger neurocircuitry are engaged (either directly or indirectly) by odor food cues even in the sated state, suggests a role for these neurons in food cue-potentiated feeding behaviors. We show that besides AgRP and POMC neurons, which are similarly activated, GHSR-containing and dopaminergic cells are also recruited by the external olfactory food-linked stimulus. One thing is certain: food cues in our society are unavoidable and any failure to adjust for even small increases in intake might lead to caloric surfeit. However, what if the *only* food available appeared to be a healthier choice, as was the case herein? These results likewise evince the importance of health nudge interventions conducted by governments worldwide, which try to steer people into healthier lifestyles and food choices [42].

## Figures and Tables

**Figure 1 nutrients-13-03101-f001:**
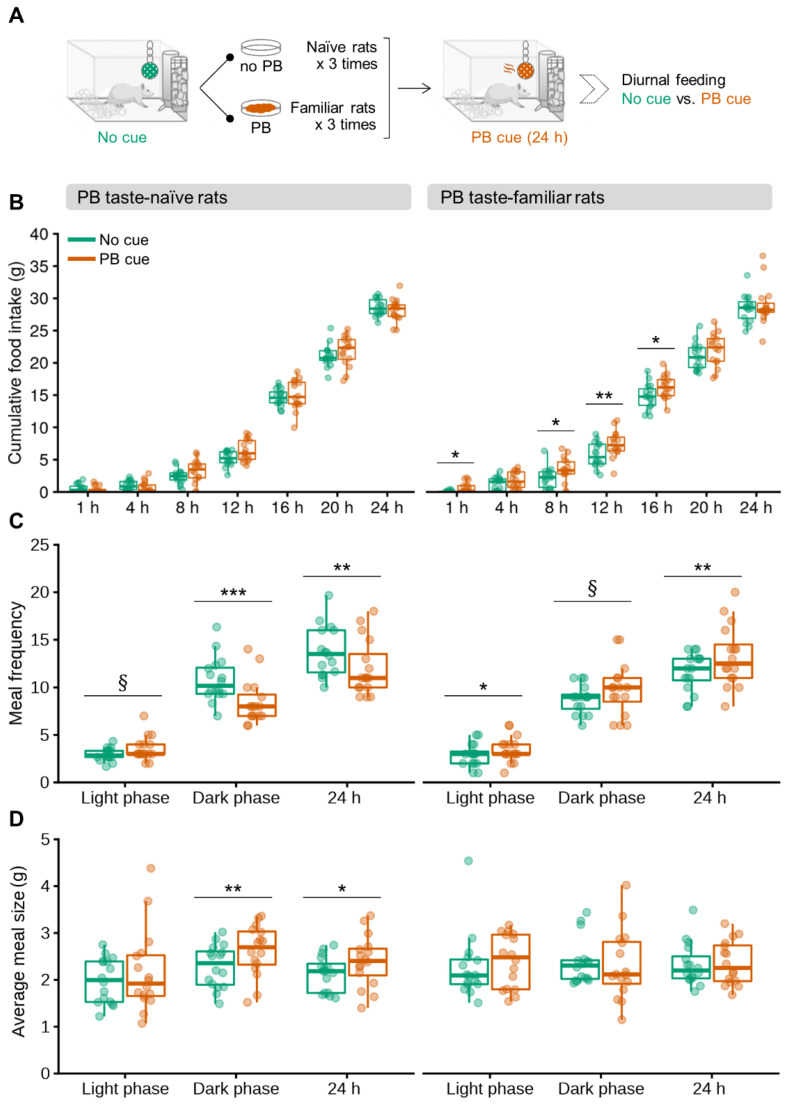
Olfactory detection of PB causes persistent overconsumption of regular chow due to an increase in meal frequency in non-food-deprived PB taste-familiar rats. (**A**) Schematic for the cross-over experimental design in the automated feeding monitoring cages; (**B**) cumulative food intake progression following introduction of the olfactory cue in the home-cage environment; (**C**) meal frequency and (**D**) average meal size (g) during the light phase, dark phase and total day (24 h) in PB taste-naïve rats (left, *n* = 16) and PB taste-familiar rats (right, *n* = 16). For illustration, the thick line always corresponds to the median, boxes show first and third quartiles and whiskers represent minimum and maximum values. Symbols indicate differences between the control and the PB cue conditions at (*) *p* < 0.05, (**) *p* < 0.01, (***) *p* < 0.001 or (§) 0.05 ≤ *p* < 0.1 by one-way repeated measures ANOVA followed by paired samples *t*-tests upon data split according to *Conditioning* (naïve, familiar). PB; peanut butter.

**Figure 2 nutrients-13-03101-f002:**
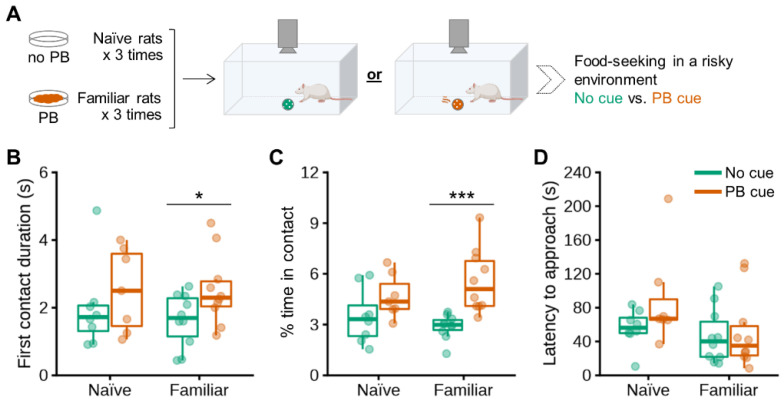
Olfactory detection of PB stimulates food-seeking in a risky environment in non-food-deprived PB taste-familiar rats. (**A**) Schematic of the experimental design in the PB-baited open field. Effects of baiting the center of the arena with PB on (**B**) first contact duration (s); (**C**) % of time in contact with the perforated ball; as well as (**D**) latency to approach the perforated ball in PB taste-naïve rats (*n* = 15: No cue, *n* = 8; PB cue, *n* = 7) and PB taste-familiar rats (*n* = 20: No cue, *n* = 10; PB cue, *n* = 10). For illustration, the thick line always corresponds to the median, boxes show first and third quartiles and whiskers represent minimum and maximum values. Symbols indicate differences between the control and the PB cue conditions at (*) *p* < 0.05 or (***) *p* < 0.001 by two-way repeated measures ANOVA with further one-way ANOVA upon data split according to *Conditioning* [naïve, familiar]. PB; peanut butter.

**Figure 3 nutrients-13-03101-f003:**
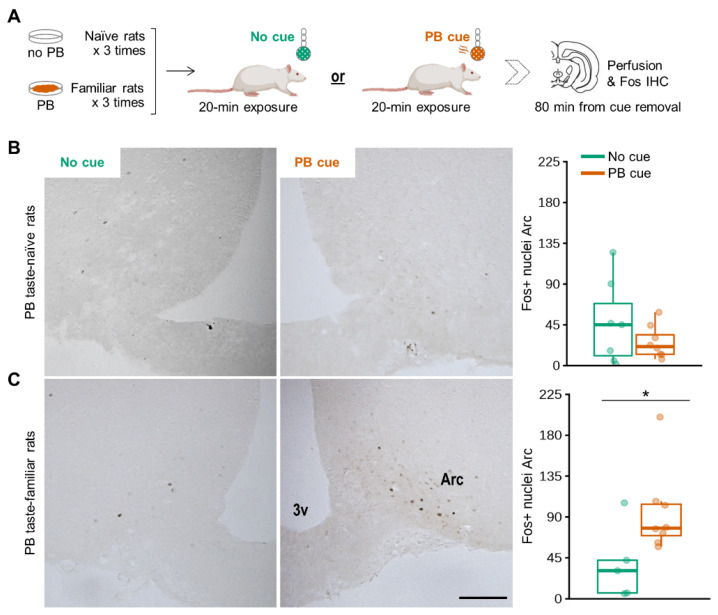
Olfactory detection of PB activates cells in the Arc in non-food-deprived PB-taste familiar rats. (**A**) Schematic of the experimental design in the home cage. Representative images showing effects of exposure to a 20-min olfactory PB cue to increase the number of cells detected that express Fos protein in the Arc, together with the corresponding unilateral manual counting of Fos+ nuclei in (**B**) PB taste-naïve rats (*n* = 15: No cue, *n* = 7; PB cue, *n* = 8) and (**C**) PB taste-familiar rats (*n* = 13: No cue, *n* = 5; PB cue, *n* = 8). Arc, arcuate nucleus; 3v, third ventricle. Bregma: −2.40 mm; scale bar = 200 μm (applies to all four images). For illustration, the thick line always corresponds to the median, boxes show first and third quartiles and whiskers represent minimum and maximum values. Symbols indicate differences between the control and the PB cue conditions at (*) *p* < 0.05 by two-way repeated measures ANOVA with further one-way ANOVA upon data split according to *Conditioning* [naïve, familiar]. PB; peanut butter.

**Figure 4 nutrients-13-03101-f004:**
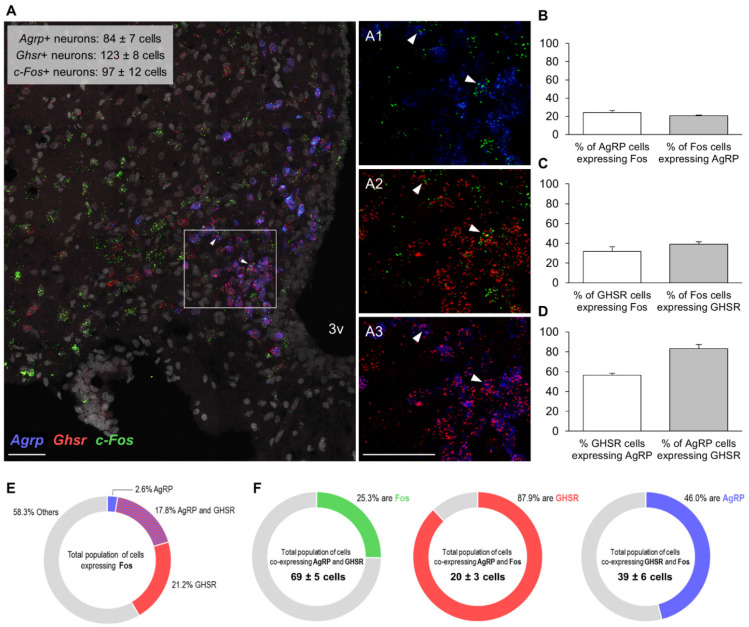
Colocalization of mRNAs for Fos protein, GHSR and AgRP in the Arc of non-food-deprived PB taste-familiar rats exposed to the olfactory PB cue. (**A**) Representative confocal images of triple RNAscope *in situ* hybridization for *c-Fos* (green), *Ghsr* (red) and *Agrp* (blue) in an Arc-containing section of a PB taste-familiar rat exposed to the olfactory PB cue. Area in the white rectangle is shown enlarged in the small panels on the right (**A1**–**A3**); colocalization of mRNAs for (**A1**) Fos protein and AgRP; (**A2**) Fos protein and GHSR; and (**A3**) GHSR and AgRP. White arrows indicate triple positive cells. The graphs depict (**B**) % of AgRP cells that are Fos+, and the % of Fos cells that are AgRP+; (**C**) % of GHSR cells that are Fos+, and the % of Fos cells that are GHSR+; (**D**) % of GHSR cells that are AgRP+, and the % of AgRP cells that are GHSR+ (Data are represented as mean ± SEM). (**E**) Overview of the molecular identities of Fos+ cells; (**F**) triple colocalization data. 3v, third ventricle. Bregma: −2.64 mm; Scale bar = 50 μm (applies to all four images). Two–four hemisections per rat were quantified (*n* = 4).

**Figure 5 nutrients-13-03101-f005:**
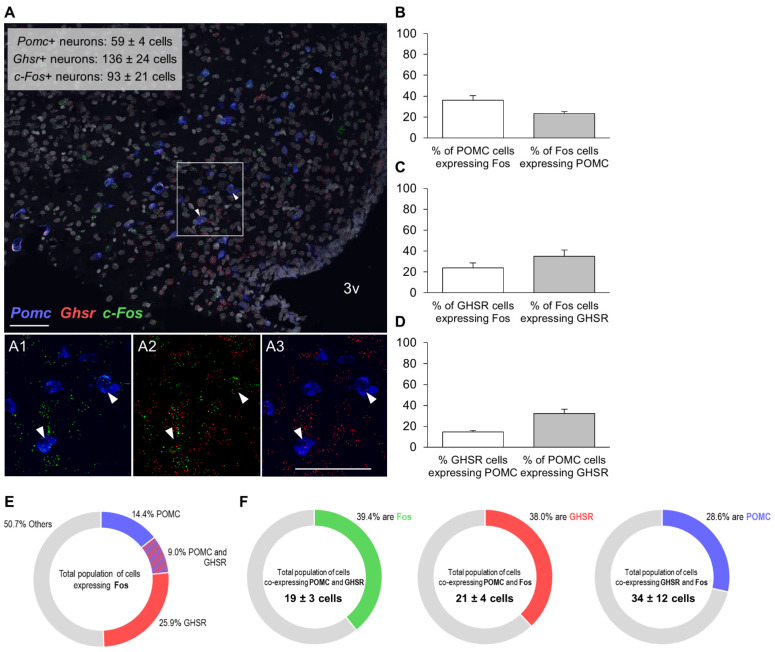
Colocalization of mRNAs for Fos protein, GHSR and POMC in the Arc of non-food-deprived PB taste-familiar rats exposed to the olfactory PB cue. (**A**) Representative confocal images of triple RNAscope *in situ* hybridization for *c-Fos* (green), *Ghsr* (red) and *Pomc* (blue) in an Arc-containing section of a PB taste-familiar rat exposed to the olfactory PB cue. Area in the white rectangle is shown enlarged in the small panels on the right (**A1**–**A3**); colocalization of mRNAs for (**A1**) Fos protein and POMC; (**A2**) Fos protein and GHSR; and (**A3**) GHSR and POMC. White arrows indicate triple positive cells. The graphs depict (**B**) % of POMC cells that are Fos+, and the % of Fos cells that are POMC+; (**C**) % of GHSR cells that are Fos+, and the % of Fos cells that are GHSR+; (**D**) % of GHSR cells that are POMC+, and the % of POMC cells that are GHSR+ (Data are represented as mean ± SEM). (**E**) Overview of the molecular identities of Fos+ cells; (**F**) triple colocalization data. 3v, third ventricle. Bregma: −3.48 mm; Scale bar = 50 μm (applies to all four images). Three–four hemisections per rat were quantified (*n* = 3).

**Figure 6 nutrients-13-03101-f006:**
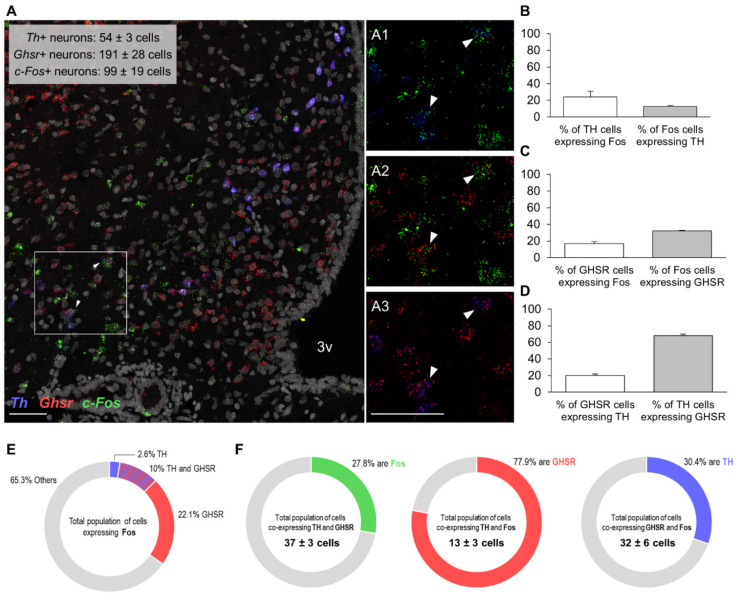
Colocalization of mRNAs for Fos protein, GHSR and TH in the Arc of non-food-deprived PB taste-familiar rats exposed to the olfactory PB cue. (**A**) Representative confocal images of triple RNAscope *in situ* hybridization for *c-Fos* (green), *Ghsr* (red) and *Th* (blue) in an Arc-containing section of a PB taste-familiar rat exposed to the olfactory PB cue. Area in the white rectangle is shown enlarged in the small panels on the right (**A1**–**A3**); colocalization of mRNAs for (**A1**) Fos protein and TH; (**A2**) Fos protein and GHSR; and (**A3**) GHSR and TH. White arrows indicate triple positive cells. The graphs depict (**B**) % of TH cells that are Fos+, and the % of Fos cells that are TH+; (**C**) % of GHSR cells that are Fos+, and the % of Fos cells that are GHSR+; (**D**) % of GHSR cells that are TH+, and the % of TH cells that are GHSR+ (Data are represented as mean ± SEM). (**E**) Overview of the molecular identities of Fos+ cells; (**F**) triple colocalization data. 3v, third ventricle. Bregma: −2.64 mm; Scale bar = 50 μm (applies to all four images). Three hemisections per rat were quantified (*n* = 3).

**Figure 7 nutrients-13-03101-f007:**
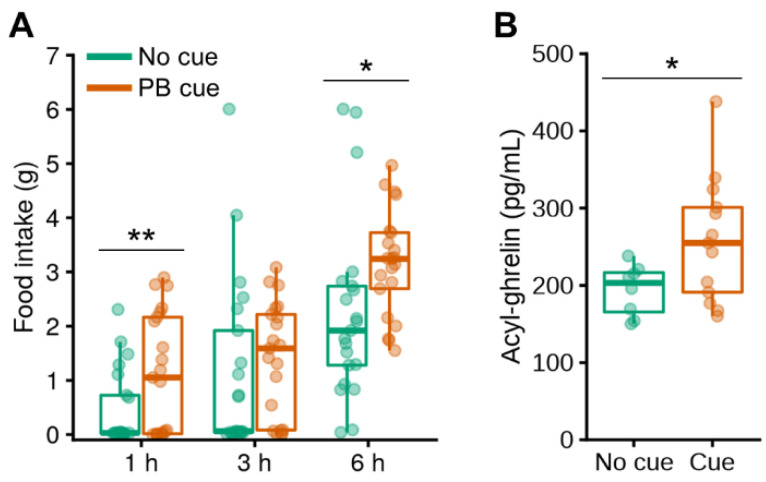
Olfactory detection of PB triggers release of active ghrelin in non-food-deprived PB taste-familiar rats. (**A**) Confirmation of the cue-induced overeating of chow in a new cohort of PB taste-familiar rats subjected to a cross-over design (*n* = 21). Manual food intake (g) measurements after 1, 3 and 6 h post-cue introduction in the home cage. The same time points were used as for the control condition on the day before, in which spontaneous food intake was measured in the absence of the PB cue following introduction of an empty tube in the perforated ball; (**B**) levels of acyl-ghrelin (also known as active ghrelin, pg/mL) under *ad libitum*-fed conditions after 1 h of exposure to the olfactory PB cue (*n* = 21: No cue, *n* = 8; PB cue, *n* = 13). For illustration, the thick line always corresponds to the median, boxes show first and third quartiles and whiskers represent minimum and maximum values. In (**A**), symbols indicate differences between the control and the PB cue conditions at (*) *p* < 0.05 or (**) *p* < 0.01 by paired samples *t*-tests; in (**B**), the symbol indicates differences between the control and the PB cue conditions at (*) *p* < 0.05 by one-way ANOVA. PB; peanut butter.

## Data Availability

The datasets generated during and/or analyzed during the present study are not publicly available, but are available from the corresponding authors upon reasonable request.

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
