# Peer review of "The Orexigenic Force of Olfactory Palatable Food Cues in Rats"

_nutrients, 2021, doi:10.3390/nu13093101_

Round 1

Reviewer 1 Report

The study intensely examined that the effect of palatable olfactory cue on feeding behavior, and provides better insights into the regulation of feeding behavior by olfactory cues and also palatable olfactory cue-induced hyperphagia.

General

1. Lines 549-551: “We show that olfactory detection of a familiar palatable food shifts diurnal feeding patterns to cause sustained overconsumption of the only available food option, here plain chow, even if the latter lacks incentive value under ad libitum-fed conditions.”

It is hardly say that olfactory cue shifts diurnal feeding patterns, because the cue was provided immediately after the onset of light phase, and the authors did not examine the effect of cue placement at the onset of dark phase, the results just show the time course after the cue presentation (as the authors discuss in lines 581-597).

2. Figure 1.

It is better to show “No cue” and “PB cue” data in Fig 1 B in a similar manner in Figs. 1C and 1D. This is because readers are able to know a circadian feeding pattern.

In addition, it is also better to show feeding every 3 or 4 hours. Because the light/dark cycle was 12/12h, the 12 h-data (the onset of dark phase) should be shown. Why did the authors set the data point at 1, 3, 6, 10, 16, and 24 h after the onset of cue access?

3. Lines 553-557: “Using Fos mapping, we identify the Arc as a key neural substrate activated by olfactory food cues and provide evidence that these recruit intermingled populations of cells embedded within the feeding circuitry, even when neither chow nor the cued PB are available for consumption.” 

The involvement of arcuate nucleus is possible, but the authors did not reveal direct connection between olfaction and feeding behavior, this sentence should be more conservative. The data did not provide evidence, but provides suggestion or possibility.

The conclusions should also be more conservative.

It is unclear why the authors conducted the mouse experiment.

Reviewer 2 Report

Overall this is a comprehensive and well-designed series of studies investigating cue-induced overeating. For example, the authors demonstrate that sensory detection of PB in the rat environment motivates the animal to eat in a manner that might mimic the state of hunger. It is also intriguing that the animals ultimately compensate for such over-consumption of energy, and as the authors suggest, homeostasis prevails. The rise in endogenous ghrelin appears to play a primary role as a key neurohormonal substrate. Further, the authors suggest that it might be the case the odor only gains value when are familiar with the taste of PB. In short, this is a very well written report that strengthens our understanding of odor food cues and their integration with homeostatic ArcN neurons. The implications with respect to healthier food choice is also intriguing.

Given the strength of the manuscript as well as the results, and data interpretation. I have no major concerns regarding the paper. However, I do have a few minor comments.

  1. It is curious that PB failed to induce c-Fos expression, particularly in reward circuitry. The author suggest that they cannot exclude brain areas where neuronal activation is not coupled to a Fos response. Could they provide an example of such a brain area/areas?
  2. The body weights reported in the Subjects section seem somewhat high for rats aged 7-9 weeks.
  3. The paper focused on male rodents. Do the authors expect similar findings in females?
  4. The authors selected energy dense PB as their cue. Would one anticipate similar responses if the cue were still pronounced as an olfactory cue, but energy deplete?

Round 2

Reviewer 1 Report

The authors revised the manuscript appropriately according to the suggestions. I have no more concerns.